# Differentially Private Federated Quantiles with the Distributed Discrete Gaussian Mechanism

**Krishna Pillutla**[*†1]   **Yassine Laguel**[*2]   **Jérôme Malick**[3]   **Zaid Harchaoui**[1]

[1] University of Washington, Seattle, WA, USA   [2] Rutgers University, New Brunswick, NJ, USA

[3] CNRS, Grenoble, France

## Abstract

The computation of analytics in a federated environment plays an increasingly important role in data science and machine learning. We consider the differentially private computation of the quantiles of a distribution of values stored on a population of clients. We present two quantile estimation algorithms based on the distributed discrete Gaussian mechanism that are compatible with secure aggregation. Based on a privacy-utility analysis and numerical experiments, we delineate the regime under which each one is superior. We find that the algorithm with suboptimal asymptotic performance works the best on moderate problem sizes typical in federated learning with client sampling. We apply these algorithms to augment distributionally robust federated learning with differential privacy.

## 1 Introduction

Federated analytics consists of a number of client devices, such as smartphones, collaboratively computing population statistics under the orchestration of a central server [19, 17]. A key component of this paradigm is the preservation of clients' privacy: any deployed algorithms must ensure that no client data is leaked through the released statistics via rigorous differential privacy guarantees [7, 8].

In the federated setting, the recent paradigm of distributed differential privacy has been growing in popularity [16]. It allows one to simulate the effect of a trusted central aggregator without actually trusting the orchestrating server, via the use of cryptographic secure aggregation protocols [2]. Practical implementations of such algorithms are based on cryptographic techniques such as secure multiparty computation [10], which add two key limitations. First, they require interactions with the server to be limited to sums of per-client vectors. Second, they require each component of the per-client vectors to be discretized to the ring $\mathbb{Z}_M$ of integers modulo $M$.

In this work, we are interested in the problem of federated quantile computation of a per-client scalar. Besides being fundamental to robust statistics, quantiles are central to adaptive clipping in federated averaging [1] and to distributionally robust versions of federated learning [18]. Prevailing approaches to differentially private quantile computation [24, 26, 13] typically require operations such as sorting or dynamic programming. It is unclear how to extend these approaches to satisfy distributed differential privacy, particularly, limiting communication with the server to sums of per-client vectors.

We study a simple alternative based on classical techniques from cumulative distribution estimation that are compatible with distributed differential privacy. We quantize the per-client scalars into a per-client histogram, which we then aggregate with a simplified variant of the distributed discrete Gaussian mechanism [16]. We show that the quantile error to obtain an $(\varepsilon, \delta)$-differentially private quantile estimate of $n$ scalars with a histogram of $b$ bins scales as $\sqrt{b}/(\varepsilon n)$ ignoring log factors.

---

[*]These authors contributed equally to this work. [†]Now at Google Research.

Workshop on Federated Learning: Recent Advances and New Challenges, in Conjunction with NeurIPS 2022 (FL-NeurIPS'22). This workshop does not have official proceedings and this paper is non-archival.

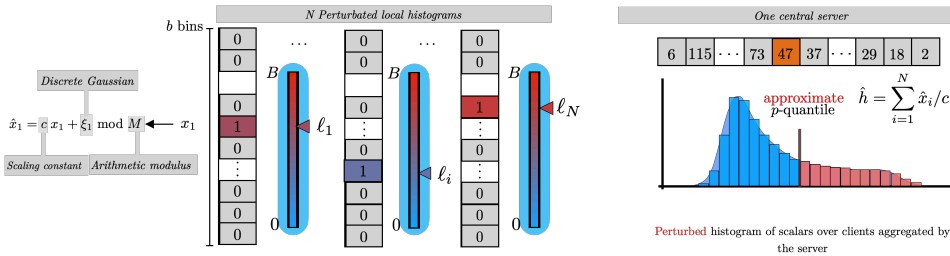

Figure 1: Illustration of our differentially private algorithm for the computation of a quantile using (flat) histograms.

Flat histograms are known to be suboptimal for cumulative distribution estimation; an approach known as the hierarchical histogram method or tree aggregation attains a better asymptotic dependence on the number of bins [15, 9, 5, 25, 6]. We show that the hierarchical histogram method with distributed differential privacy can attain a quantile error of $\log^2 b/(\varepsilon n)$.

While the hierarchical method attains a better asymptotic dependence on the number $b$ of bins, the flat method achieves a smaller error for moderate $b$, as might be typical in a federated learning setting due to client sampling [17]. We corroborate this observation with synthetic numerical experiments in quantile estimation and distributionally robust federated learning.

## 2 Setting and Algorithms

Suppose we have $n$ clients, with each client $i$ hosting a bounded scalar $\ell_i \in [0, B], i \in \{1, \cdots, n\}$ for a fixed $B > 0$. We wish to estimate the $p$-quantile of the empirical distribution of $\ell_i$'s, defined as

$$q_p := \min \left\{ t \, : \, \frac{\sum_{i=1}^n \mathbb{I}(\ell_i \leq t)}{n} \leq p \right\} \, .$$

Distributed differential privacy simulates a trusted central aggregator by using a secure summation oracle [2], which enables the computation of summations $\sum_{i=1}^n v_i$ where $v_i \in \mathbb{R}^d$ is a privacy-sensitive vector residing with client $i$ without revealing any further information to a privacy adversary. Practical implementations of such algorithms are based on cryptographic techniques such as secure multiparty computation [10], which requires each component of the vectors $v_i$ to be discretized to the ring $\mathbb{Z}_M$ of integers modulo $M$.

**Quantiles from Flat Histograms.** We start with by estimating the quantile from a flat histogram, as illustrated in Figure 1. Each client $i$ first computes a local histogram $x_i \in \{0, 1\}^b$ on $[0, B]$ with $b$ bins and given edges $0 \leq l_0 < l_1 < \cdots < l_b = B$. Each client then adds random discrete Gaussian with noise $\xi_i \sim \mathcal{N}_{\mathbb{Z}}(0, \sigma^2 I_b)$ and scale parameter $\sigma^2$, that is a random variable $\xi$ over $\mathbb{Z}$ satisfying

$$\mathbb{P}(\xi = i) = C \, \exp\left(-\frac{i^2}{2\sigma^2}\right) \quad \text{for all } i \in \mathbb{Z} \, ,$$

where $C$ is an appropriate normalizing constant. We finally sum them up using a secure summation oracle. At the end of all these steps, the server has a histogram $\hat{h} \in \mathbb{R}^b$ which approximates the true histogram $h = \sum_{i=1}^n x_i$ of per-client scalars. The cumulative distribution $\hat{F}_j = (\hat{h}_1 + \cdots + \hat{h}_j)/(\hat{h}_1 + \cdots + \hat{h}_b)$ induced by the approximate histogram $\hat{h}$ estimates the true cumulative distribution $F_j = (1/n) \sum_{i=1}^n \mathbb{I}(\ell_i \leq l_j)$. Since it may fail to be monotonic, we estimate the quantile using a bin edge $l_j$ such that the estimated cumulative mass $\hat{F}_j$ is as close to $p$ as possible:

$$Q_p(\hat{F}) := l_{j_p^*(\hat{F})} \quad \text{where} \quad j_p^*(\hat{F}) \quad = \arg \min_{j \in [b]} |\hat{F}_j - p| \, . \tag{1}$$

The full algorithm is given in Algorithm 1. We also consider a variant of the algorithm where the cumulative distribution $\hat{F}_j$ is estimated using the exact counts, i.e., the denominator $\hat{h}_1 + \cdots + \hat{h}_b$ is replaced by the true sum $h_1 + \cdots + h_b = n$ (option 2 in Algorithm 1).

---

**Algorithm 1** Quantile Computation with Distributed Differential Privacy: Flat Histograms

---

**Input:** Ring size $M$, $n$ clients each with $\ell_i \in [0, B]$, target quantile $p \in (0, 1)$, discretization $l_0, l_1, \cdots, l_b$ of $[0, B]$, variance proxy $\sigma^2$, scaling factor $c \in \mathbb{Z}_+$

1: Each client $i$ computes local histogram $x_i = \left(\mathbb{I}(l_{j-1} \leq \ell_i < l_j)\right)_{j=1}^{b}$
2: Each client $i$ samples $\xi_i \sim \mathcal{N}_\mathbb{Z}(0, \sigma^2 I_b)$ and sets $\tilde{x}_i = (cx_i + \xi_i) \mod M$
3: Compute $s = (\sum_{i=1}^{n} \tilde{x}_i) \mod M$ securely and set histogram $\hat{h} = s/c$
4: Estimate the cumulative function with $\hat{F} \in \mathbb{R}^b$ as
   - **Option 1 (estimated count):** $\hat{F}_j = (\hat{h}_1 + \cdots + \hat{h}_j)/\hat{n}$ where $\hat{n} = \hat{h}_1 + \cdots + \hat{h}_b$
   - **Option 2 (exact count):** $\hat{F}_j = (\hat{h}_1 + \cdots + \hat{h}_j)/n$
5: **return** Quantile estimate $l_{j_p^*(\hat{h})}$ corresponding to index $j_p^*(\hat{h})$; cf. Eq. (1)

---

**Quantiles from Hierarchical Histograms.** A hierarchical histogram $H$ maintains the number of clients not only in every single bin but also in groups of bins organized as a binary tree. Concretely, $H(r, j)$ maintains the number of clients whose losses lie between the bin edges $l_{2^r(j-1)+1}$ and $l_{2^r j}$ for index $j = 1, \cdots, b/2^r$ and level $r = 0, \cdots, \log_2 b - 1$.[1] The lower levels $r = 0$ and $r = 1$ correspond respectively to individual bins and pairs of bins, while the topmost level $r = \log_2 b - 1$ refers to two groups: the first $b/2$ bins and the last $b/2$ bins. We skip the topmost level in the tree because the count at this node is the publicly known number $n$ of clients.

Each client $i$ first computes its local hierarchical histogram $X_i$ as

$$X_i(r, j) = \mathbb{I}\left(l_{2^r(j-1)+1} \leq \ell_i < l_{2^r j}\right) ,$$

such that the overall hierarchical histogram can be obtained as $H = \sum_{i=1}^{n} X_i$. As previously, each client adds discrete Gaussian noise $\xi_i \sim \mathcal{N}_\mathbb{Z}(0, \sigma^2 I)$ with scale parameter $\sigma^2$. These noisy hierarchical histograms are summed up using a secure summation oracle so that the server receives an approximate hierarchical histogram $\hat{H}$ which approximates the true $H = \sum_{i=1}^{n} X_i'$.

The benefit of hierarchical histograms for cumulative distribution estimation comes from utilizing counts at nodes higher up in the tree. Using a maximal dyadic partition $P_j$ of the range $[1, j]$, we have $F(j) = \sum_{(r,o) \in P_j} H(r, o)$ from summing up $|P_j| \leq \log_2 b$ terms. For instance, the dyadic partition for $j = 15$ is $[1, 15] = [1, 8] \cup [9, 12] \cup [13, 14] \cup [15]$, where the counts of each range on the right side can be obtained from an intermediate node in the hierarchical histogram $H$.

Similarly, we estimate the cumulative distribution from the approximate hierarchical histogram $\hat{H}$ as $\hat{F}_j = \sum_{(r,o) \in P_j} \hat{H}(r, o)$ where $P_j$ is a maximal dyadic partition of $[1, j]$. Similar to the previous approach, we estimate the quantile with (1). The full algorithm is given in Algorithm 2.

## 3 Utility and Privacy Analysis

We now analyze the privacy and utility of Algorithm 1. First, recall the definition of zero-concentrated differential privacy [4]: a randomized algorithm $A$ satisfies $(1/2)\varepsilon^2$-concentrated differential privacy if the Rényi $\alpha$-divergence $D_\alpha(A(\boldsymbol{x})\|A(\boldsymbol{x}')) \leq \alpha\varepsilon^2/2$ for all $\alpha \in (0, \infty)$ and all sequences $\boldsymbol{x}, \boldsymbol{x}'$ of inputs which differ by the addition or removal of the data of one client. Intuitively, the addition or removal of one client should not change the output distribution of the randomized algorithm by much, as measured by the Rényi divergence. A smaller value of $\varepsilon$ implies a stronger privacy guarantee.

**Error Criterion.** The bin edge $l_j$ corresponding to index $j \in [b]$ approximates the $p$-quantile well if the cumulative mass $F_j \approx p$. We measure this error of approximation by the difference between the two sides. Formally, we define the error $R_p(F, j)$ of approximating the $p$-quantile of the cumulative function $F$ of a (hierarchical) histogram with index $j \in [b]$ by

$$R_p(F, j) = |F_j - p| . \tag{2}$$

---

[1]We assume for simplicity that $b$ is a power of 2 so that $\log_2 b$ is an integer.

**Algorithm 2** Quantile Computation with Distributed Differential Privacy: Hierarchical Histograms

---

**Input:** Ring size $M$, $n$ clients each with $\ell_i \in [0, B]$, target quantile $p \in (0, 1)$, discretization $l_0, l_1, \cdots, l_b$ of $[0, B]$, variance proxy $\sigma^2$, scaling factor $c \in \mathbb{Z}_+$
 1: Each client $i$ computes a hierarchical histogram $X_i(r, j) = \mathbb{I}\left(l_{2^r(j-1)+1} \le \ell_i < l_{2^r j}\right)$ for $j = 1, \cdots, b/2^r$ and $r = 0, \cdots, \log_2 b - 1$
 2: Each client $i$ samples $\xi_i(r, j) \sim \mathcal{N}_{\mathbb{Z}}(0, \sigma^2)$ i.i.d. and sets $\tilde{X}_i(r, j) = \left(cX_i(r, j) + \xi_i(r, j)\right)$ mod $M$ for each $r, j$
 3: Compute $S = \left(\sum_{i=1}^n \tilde{X}_i\right) \mod M$ securely and set $\hat{H} = S/c$
 4: Define the cumulative distribution $\hat{F} \in \mathbb{R}^b$ as $\hat{F}(j) = (1/n) \sum_{(r,o) \in P_j} \hat{H}(r, o)$ using a maximal dyadic partition $P_j$ of $[1, j]$
 5: **return** Quantile estimate $l_{j_p^*(\hat{F})}$ corresponding to index $j_p^*(\hat{F})$; cf. Eq. (1)

---

We define the best achievable error a $R_p^*(F) = \min_{j \in [b]} R_p(F, j)$. Note that the best index $j_p^*(F)$ for estimating the $p$-quantile of the cumulative function $F$ as defined in (1) satisfies $j_p^*(F) = \arg \min_{j \in [b]} R_p(F, j)$. Lastly, we define the *quantile error* $\Delta_p(F, \hat{F})$ of estimating the $p$-quantile of the cumulative function $F$ from that of $\hat{F}$ as

$$\Delta_p(\hat{F}, F) = R_p\left(F, j_p^*(\hat{F})\right). \tag{3}$$

Essentially, if the index $j_p^*(\hat{F})$ computed from the estimate $\hat{F}$ corresponds to the $p'$-quantile of $F$, the quantile error satisfies $\Delta_p(\hat{F}, F) = |p - p'|$.

**Privacy and Utility Analysis.** We now analyze the differential privacy bound of the flat histogram approach of Algorithm 1 as well as the error in the quantile computation.

**Theorem 1** (Flat histogram). *Fix a $\delta > 0$. Suppose that $\sigma \ge 1/2$ and $c > 0$ are given. We have that Algorithm 1 satisfies $(1/2)\varepsilon^2$-concentrated DP with*

$$\varepsilon = \min\left\{\sqrt{\frac{c^2}{n\sigma^2} + \frac{\psi b}{2}}, \frac{c}{\sqrt{n}\sigma} + \psi\sqrt{b}\right\},$$

*where $\psi = 10 \sum_{i=1}^{n-1} \exp\left(-2\pi^2\sigma^2 i/(i+1)\right) \le 10(n-1)\exp(-2\pi^2\sigma^2)$. Further, if the modular base satisfies $M \ge 2 + 2cn + 2n\sqrt{2\sigma^2 \log(8nb/\delta)}$, then we have with probability at least $1 - \delta$ that the quantile error of cumulative function $\hat{F}$ returned by Algorithm 1 (option 2) is at most*

$$\Delta_p(\hat{F}, F) \le R_p^*(\hat{F}) + \sqrt{\frac{2\sigma^2 b}{c^2 n} \log \frac{4}{\delta}},$$

*where $R_p^*(\hat{F})$ is the error in the estimation of $p$-quantile of histogram $\hat{F}$.*

Let us interpret the result. The effective noise scale is $\sigma/c$. Since the dominant term of the privacy error is $\varepsilon \approx c/(\sigma\sqrt{n})$, we choose $\sigma/c \approx (\varepsilon\sqrt{n})^{-1}$, so that the algorithm satisfies $(1/2)\varepsilon^2$-concentrated DP. The role of $c$ is to avoid degeneracy of the discrete Gaussian as $\sigma \to 0$. In particular, the theorem requires $\sigma \ge 1/2$. Ignoring constants and log factors, the error is

$$\Delta_p(\hat{F}, F) \lesssim R_p^*(\hat{F}) + \frac{\sqrt{b}}{\varepsilon n}.$$

While the quantile error increases with the number $b$ of bins, the discretization error from reducing the scalars to a histogram typically reduces with $b$. For instance, for a uniform discretization of $[0, B]$, the $p$-quantile of the histogram $\sum_{i=1}^n x_i$ is at most $B/b$ away from the true $p$-quantile of the scalars $\ell_1, \cdots, \ell_n$. Finally, if we take $\sigma = O(1)$ and $c = O(\varepsilon\sqrt{n})$, the conditions of the theorem require $M \gtrsim n^{3/2}$. We give a similar bound for Algorithm 1 with estimated $n$ (option 1) in Appendix A.

**Theorem 2** (Hierarchical Histogram). *Fix a $\delta > 0$. Suppose that $\sigma \ge 1/2$ and $c > 0$ are given. We have that Algorithm 2 satisfies $(1/2)\varepsilon^2$-concentrated DP with ($\psi$ is given in Theorem 1)*

$$\varepsilon = \min\left\{\sqrt{\frac{c^2 \log_2^2 b}{n\sigma^2} + \psi b}, \frac{c \log_2 b}{\sqrt{n}\sigma} + \psi\sqrt{2b}\right\}.$$

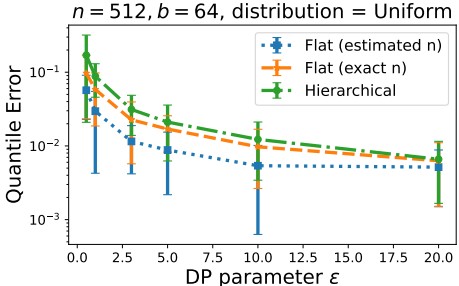 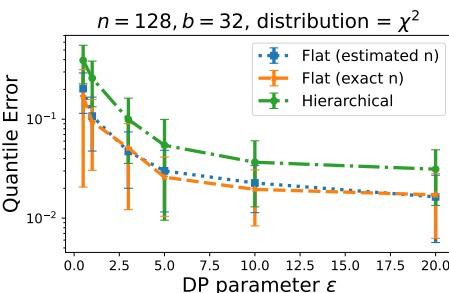

Figure 2: Distributed quantile estimation with Algorithms 1 and 2. Here, $n$ is the number of clients, $b$ is the number of bins, and bit width refers to $\log_2 M$ where $M$ is the ring size. The scalar in each client is drawn from either $\text{Unif}([0, B])$ or $\chi^2(4)$ distribution clipped to $[0, B]$, where $B = 10$ in both cases. We plot the worst quantile error (3) in computing the $p$-quantile for $p \in \{0.1, 0.2, \cdots, 0.9\}$, versus $\varepsilon$ for $(\varepsilon, 10^{-5})$-DP. We plot the mean values across 10 runs, while the error bars denote the standard deviation.

*Further, if the modular arithmetic is performed on base $M \geq 2 + 2cn + 2n\sqrt{2\sigma^2 \log(16nb/\delta)}$, then we have with probability at least $1 - \delta$ that the quantile error of cumulative function $\hat{F}$ returned by Algorithm 2 is at most*

$$\Delta_p(\hat{F}, F) \leq R_p^*(\hat{F}) + \sqrt{\frac{4\sigma^2}{c^2 n} \log_2 b \, \log \frac{4b}{\delta}} \, .$$

**Comparing the Two Approaches.** Asymptotically, the flat histogram approach has an error of $\sqrt{b}/\varepsilon n$, while the hierarchical one has an error of $\text{poly} \log b/\varepsilon n$, showing the asymptotic optimality (up to log factors) of the latter. However, the behavior is more nuanced for intermediate values of $b$ and $n$. In particular, disregarding the error term $\psi$ in the privacy cost, we have that the flat approach is better for

$$\frac{\sqrt{2b \log(4/\delta)}}{\varepsilon n} \leq \frac{\sqrt{4 \log_2^3(b) \, \log(4b/\delta)}}{\varepsilon n} \, .$$

With a failure probability of $\delta = 0.05$, this corresponds (approximately) to $b \leq 2000$. Under the Freedman–Diaconis rule [12] where one chooses the number of bins as $b = O(n^{2/3})$, the flat histogram is better for $n$ approximately below $2.5 \times 10^6$. In federated learning scenarios where quantile statistics are computed along with learning (e.g. [1, 18]), $n$ is the number of clients per round and is of the order of $10^2$ to $10^4$ in typical cross-device settings [17]. Our analysis suggests that we should prefer the flat histogram approach in this case although it is asymptotically suboptimal.

## 4 Numerical Experiments: Quantile Estimation

We test the privacy and utility of Algorithms 1 and 2 in synthetic examples. We consider $n \in \{128, 512\}$ clients, each with a scalar that is distributed as $\text{Unif}([0, B])$ or $\chi^2(4)$ distributions. We clip all scalars to $[0, B]$ with $B = 10$. For all experiments, we consider the uniform quantization of $[0, B]$ into $b$ bins, where $b \in \{32, 64\}$. We fix the ring size at $M = 2^{18}$, which is large enough to prevent any modular wraparound as required by the theorems.

Both Algorithms 1 and 2 require two further parameters: the variance-proxy $\sigma^2$ and the granularity parameter $c$, which together determine the privacy leakage. As per the discussion following Theorem 1, we take $\sigma^2 = 2$ and vary $c$. We plot the utility of each algorithm, as measured by the quantile error (3) versus the privacy parameter $\varepsilon$ required for $(\varepsilon, 10^{-5})$-differential privacy.

**Results.** Figure 2 shows the numerical results. First, we note that all three approaches have small quantile errors. For the left plot, the flat histogram with estimated $n$ (Algorithm 1, option 1) has a quantile error of 0.03 at $\varepsilon = 1$; this means we might find the 47th or 53rd percentile instead of the median. This error quickly falls below 0.01 at $\varepsilon = 5$.

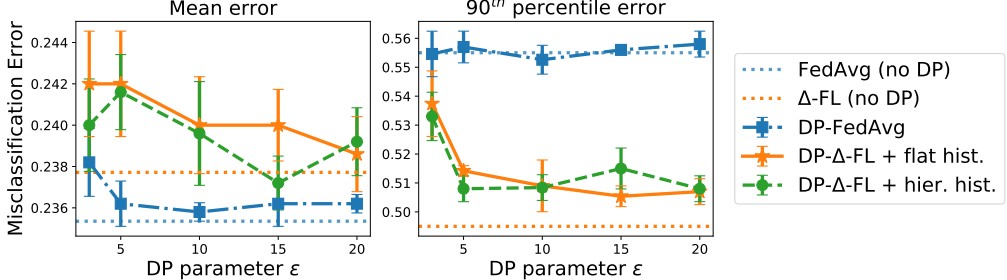

Figure 3: $\Delta$-FL vs. FedAvg with $(\varepsilon, 1/n)$-DP on a synthetic classification task in $\mathbb{R}^{20}$ with 10 classes and $n = 2500$ clients and a tail parameter $p = 0.75$. The error bars denote the standard deviation across 5 runs.

Second, we note that the flat histogram approaches tend to outperform the hierarchical approach at small $\varepsilon$ despite their asymptotic suboptimality. For instance, the flat histogram with estimated $n$ (error = 0.03) vastly outperforms the hierarchical approach (error = 0.09) at $\varepsilon = 1$ in the left plot. These values are 0.10 and 0.26 respectively for the right plot; the latter might be too inexact to even be useful depending on the application. At $\varepsilon \geq 10$, all the methods are within one standard deviation of each other. Overall, these results corroborate the theoretical analysis of Section 3.

## 5 Application to Distributionally Robust Federated Learning

We develop an end-to-end differentially private version of $\Delta$-FL [18], a distributionally robust approach to federated learning. It minimizes the tail mean of the per-client losses, known formally as the superquantile [23], which is defined for a continuous random variable $Z$ as $\mathbb{S}_p(Z) = \mathbb{E}[Z|Z > Q_p(Z)]$, where $Q_p(Z)$ is its $p$-quantile. In each round of their algorithm consists of two steps:

(a) **Quantile estimation**: Compute the $p$-quantile of the per-client losses, and,

(b) **Tail Aggregation**: Aggregate local updates from clients whose loss is larger than the quantile.

We propose an end-to-end differentially private version of this algorithm, called DP-$\Delta$-FL. We estimate the quantile of the losses clipped to a tuned bound $B$ using Algorithm 1 or 2. We clip the weight updates to a norm bound $C$, and add Gaussian noise, similar to DP-FedAvg [20]. The total privacy loss is calculated by composing the privacy loss across both the quantile and weight updates, and the number of rounds together with amplification by subsampling using the bounds of [27].

We calculate the noise scales $\sigma_q$ of the quantile and $\sigma_w$ of the weight update so that the overall algorithm satisfies $(\varepsilon, 1/n)$-differential privacy and tune the loss bound $B$, norm bound $C$, the number of bins $b$ and the ratio of the privacy budget to assign to the quantile estimation.

We experiment with a synthetic dataset contains $k = 10$ classes in $d = 20$ dimensions, and $n = 2500$ training clients, each with 100 examples. We consider robustness to a label shift, while the class-conditional distribution is fixed. The marginal across classes is a Dirichlet distribution with parameter 0.5 for the training clients and 0.01 for the test clients. For further details, we refer to Appendix B.

**Results.** We compare the distributional robustness in terms of the 90th percentile of the per-client errors in Figure 3; we also show the mean test error. First, we see that both the flat or hierarchical histograms lead to similar performance for DP-$\Delta$-FL (equal up to one std.).

We also observe that DP-$\Delta$-FL has a much smaller tail error than DP-FedAvg *at each* $\varepsilon$; the gap is 4.3 pp at $\varepsilon = 5$. Its performance does not degrade much with privacy and is within 1pp of $\Delta$-FL without DP at $\varepsilon = 5$. Importantly, *the 90th percentile error of DP-$\Delta$-FL is better than FedAvg without DP*. In terms of the mean error, DP-$\Delta$-FL is within 0.2-0.4 pp to DP-FedAvg, comparable to the gap between the non-private versions (0.24 pp).

We conclude from the experiments that (a) DP-$\Delta$-FL has a privacy-utility tradeoff similar to DP-FedAvg compared to their respective non-private versions, and (b) the distributional robustness of $\Delta$-FL is also enjoyed by its private variant.

**Acknowledgements**

The authors thank Peter Kairouz and Lun Wang for fruitful discussions. We acknowledge support from NSF DMS 2023166, DMS 1839371, CCF 2019844, the CIFAR program "Learning in Machines and Brains", faculty research awards, and a JP Morgan PhD fellowship. This work has been partially supported by MIAI – Grenoble Alpes, (ANR-19-P3IA-0003). This work was performed while Krishna Pillutla was at the University of Washington.

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

## A  Additional Results and Proof Details

In this section, we give some additional results and present the proofs of the privacy-utility bounds.

For ease of handling negative integers, we perform modular arithmetic throughout over the ring $\{-M/2 + 1, \cdots, -1, 0, 1, 2, \cdots, M/2\}$ rather than $\{0, 1, \cdots, M - 1\}$.

**Flat Histogram with Estimated Counts.** We first give the analogue of Theorem 1 for the option of estimating the counts.

**Theorem 3.** *Consider the setting of Theorem 1. If the modular base satisfies $M \geq 2 + 2cn + 2n\sqrt{2\sigma^2 \log(8nb/\delta)}$, then we have with probability at least $1 - \delta$ that the quantile error of cumulative function $\hat{F}$ returned by Algorithm 1 (option 1, estimated count) is at most*

$$\Delta_p(\hat{F}, F) \leq R_p^*(\hat{F}) \left(1 + \sqrt{\frac{2\sigma^2 b}{c^2 n} \log \frac{4}{\delta}}\right) + (1 + p)\sqrt{\frac{2\sigma^2 b}{c^2 n} \log \frac{4}{\delta}},$$

*where $R_p^*(\hat{F})$ is the error in the estimation of $p$-quantile of histogram $\hat{F}$.*

Compared to Theorem 1, this only has an additional factor on the irreducible error term $R_p^*(\hat{F})$.

### A.1 Privacy-Utility with Flat Histograms

We now prove Theorem 1 and Theorem 3.

*Proof of Theorem 1 and Theorem 3.* We start by defining and controlling the probabilities of some events. Throughout, let $\delta > 0$ be fixed. Define the event

$$E_{\mathrm{mod}} = \bigcap_{i=1}^{n} \bigcap_{j=1}^{b} \left\{ -\frac{M-2}{2n} \leq cx_{i,j} + \xi_{i,j} \leq \frac{M-2}{2n} \right\}. \tag{4}$$

Note that under $E_{\mathrm{mod}}$, no modular wraparound occurs in the algorithm, i.e., $\tilde{x}_i = cx_i + \xi_i$ and

$$\hat{h} = \sum_{i=1}^{n} \frac{\tilde{x}_k}{c} = \sum_{i=1}^{n} \left( x_k + \frac{\xi_k}{c} \right).$$

We will show later that $E_{\mathrm{mod}}$ holds with high probability; for now, we assume that it holds.

**Privacy Analysis.** We start by establishing the sensitivity of the sum query over $x_k$'s as 1. Define the input space $\mathcal{X}$ to be the canonical basis vectors in $\mathbb{R}^b$, i.e., the set of all vectors in $\{0, 1\}^b$ with only one 1, and let $\mathcal{X}^* = \cup_{r=1}^{\infty} \mathcal{X}^r$ denote the set of all sequences of elements of $\mathcal{X}$. We consider the rescaled sum query $A((x_1, \cdots, x_N)) = \sum_{i=1}^{n} cx_i$. The $L_2$ sensitivity $S(A)$ of this query $A$ is supremum over all $X \in \mathcal{X}^*$ and $X'$ which is obtained by concatenating $x'$ to $X$:

$$S(A) = \sup_{X, X'} \|A(X) - A(X')\|_2 = \sup_{x' \in \mathcal{X}} c \|x'\|_2 = c.$$

We invoke the privacy bound of sums of discrete Gaussians (see e.g. [16]) to claim that an algorithm $\mathcal{A}$ returning $A(x) + \sum_{i=1}^{n} \xi_k$ satisfies $(1/2)\varepsilon^2$-concentrated DP where $\varepsilon$ is as in the theorem statement. The fact that the quantile and all further functions of it remains private follows from the post-processing property of DP (also known as the data-processing inequality).

**Utility Analysis with Option 1 (estimated count, Theorem 3).** Define $\hat{n} = \sum_{j=1}^{b} \hat{h}_j$, as the noisy analogue to $n = \sum_{j=1}^{b} h_j$. We bound the quantile error as

$$\Delta_p\left(\hat{F}, F\right) = R_p\left(F, j_p^*(\hat{F})\right) = \left| \frac{1}{n} \sum_{j=1}^{j_p^*(\hat{F})} h_j - p \right|$$

$$\leq \frac{1}{n} \left| \sum_{j=1}^{j_p^*(\hat{F})} h_j - \hat{h}_j \right| + \frac{1}{n} \left| \sum_{j=1}^{j_p^*(\hat{F})} \hat{h}_j - \hat{n}p \right| + \frac{p}{n} |\hat{n} - n|$$

$$\leq \max_{j' \in [b]} \frac{1}{cn} \left| \sum_{j=1}^{j'} \sum_{i=1}^{n} \xi_{i,j} \right| + \left(1 + \frac{|\hat{n} - n|}{n}\right) R_p^*(\hat{F}) + \frac{p}{n} |\hat{n} - n|.$$

Let us define an event $E_{\mathrm{sum}}$ under which the first term and last terms are bounded:

$$E_{\mathrm{sum}} = \left\{ \max_{j \in [b]} \left| \sum_{j'=1}^{j} \sum_{i=1}^{n} \xi_{i,j'} \right| \leq \sqrt{2\sigma^2 nb \log(4/\delta)} \right\}. \tag{5}$$

Under $E_{\text{sum}}$, we also have

$$|n - \hat{n}| = \frac{1}{c} \left| \sum_{j=1}^{b} \sum_{i=1}^{n} \xi_{i,j} \right| \leq \sqrt{\frac{2\sigma^2 nb}{c^2} \log \frac{4}{\delta}}.$$

Plugging this back into $\Delta_p(\hat{h}, h)$ gives us the desired bound, provided $E_{\text{sum}}$ holds.

**Utility Analysis with Option 2 (exact count, Theorem 1).** Similar to the previous case, we have

$$\Delta_p \left( \hat{F}, F \right) = R_p \left( F, j_p^*(\hat{F}) \right) = \left| \frac{1}{n} \sum_{j=1}^{j_p^*(\hat{F})} h_j - p \right|$$

$$\leq \frac{1}{n} \left| \sum_{j=1}^{j_p^*(\hat{F})} h_j - \hat{h}_j \right| + \left| \frac{1}{n} \sum_{j=1}^{j_p^*(\hat{F})} \hat{h}_j - p \right|$$

$$\leq \max_{j' \in [b]} \frac{1}{cn} \left| \sum_{j=1}^{j'} \sum_{i=1}^{n} \xi_{i,j} \right| + R_p^*(\hat{F}).$$

Under $E_{\text{sum}}$, the desired bound holds.

**Bounding the Failure Probability.** The algorithm fails when at least one of $E_{\text{mod}}$ or $E_{\text{sum}}$ fail to hold. We have from Claim 4 that $\mathbb{P}(E_{\text{mod}}) \geq 1 - \delta/4$ under the given assumptions. From, Claim 5, we have $\mathbb{P}(E_{\text{sum}}|E_{\text{mod}}) \geq 1 - \delta/2$. We bound the total failure probability of the algorithm with a union bound as

$$\mathbb{P}(\bar{E}_{\text{sum}} \cup \bar{E}_{\text{mod}}) \leq \mathbb{P}(\bar{E}_{\text{sum}}|E_{\text{mod}}) \mathbb{P}(E_{\text{mod}}) + \mathbb{P}(\bar{E}_{\text{sum}}|\bar{E}_{\text{mod}}) \mathbb{P}(\bar{E}_{\text{mod}}) + \mathbb{P}(\bar{E}_{\text{mod}})$$

$$\leq \mathbb{P}(\bar{E}_{\text{sum}}|E_{\text{mod}}) + 2 \mathbb{P}(\bar{E}_{\text{mod}}) \leq \delta.$$

$\square$

We state and prove bounds on probabilities of the events $E_{\text{mod}}, E_{\text{sum}}$ defined above.

**Claim 4.** *If $M \geq 2 + 2cn + 2n\sqrt{2\sigma^2 \log(8nb/\delta)}$, then $\mathbb{P}(E_{\text{mod}}) \geq 1 - \delta/4$.*

*Proof.* Each discrete Gaussian random variable $\xi_{i,j}$ is centered and sub-Gaussian with variance proxy $\sigma^2$. A Cramér-Chernoff bound (see e.g. Lemma 10) gives us the exponential tail bound

$$\mathbb{P}\left( |\xi_{i,j}| > \sqrt{2\sigma^2 \log(8nb/\delta)} \right) \leq \frac{\delta}{4nb}.$$

Using a union bound for $i \in [n], j \in [b]$ and $x_{i,j} \in \{0, 1\}$ completes the proof. $\square$

**Claim 5.** *We have that $\mathbb{P}(E_{\text{sum}}|E_{\text{mod}}) \geq 1 - \delta/2$.*

*Proof.* Under $E_{\text{mod}}$, no modular wraparound occurs, so we can ignore the modulo operations. Each discrete Gaussian random variable $\xi_{i,j}$ is centered and sub-Gaussian with variance proxy $\sigma^2$, i.e., $\mathbb{E}[\xi_{i,j}] = 0$ and $\mathbb{E}[\exp(\lambda \xi_{i,j})] \leq \exp(\lambda^2 \sigma^2/2)$ for all $\lambda \in \mathbb{R}$. Therefore, $\zeta_j := \sum_{i=1}^{n} \xi_{i,j}$ is centered and sub-Gaussian with variance proxy $n\sigma^2$, since $\mathbb{E}[\zeta_j] = 0$, and

$$\mathbb{E}[\exp(\lambda \zeta_j)] = \prod_{i=1}^{n} \mathbb{E}[\exp(\lambda \xi_{i,j})] \leq \exp(\lambda^2 \sigma^2 n/2)$$

by independence. We get a bound on the partial sums from the maximal inequality (see e.g. Lemma 11); this involves constructing a martingale $(\sum_{j'=1}^{j} \zeta_{j'})_{j=1}^{b}$ and applying the maximal inequality. The bound we get is

$$\mathbb{P}\left( \max_{j \in [b]} \left| \sum_{j'=1}^{j} \zeta_{j'} \right| > t \right) \leq \exp\left( -\frac{t^2}{2\sigma^2 nb} \right).$$

Plugging in $t = \sqrt{2\sigma^2 nb \log(2/\delta)}$ completes the proof. $\square$

## A.2 Privacy-Utility with Hierarchical Histograms

We now prove Theorem 2.

*Proof of Theorem 2.* We start by defining and controlling the probabilities of some events. Throughout, let $\delta > 0$ be fixed. Define the event

$$E_{\text{mod}} = \bigcap_{i=1}^{n} \bigcap_{r=0}^{\log_2 b - 1} \bigcap_{j=1}^{b/2^r} \left\{ -\frac{M-2}{2n} \le cX_i(r,j) + \xi_i(r,j) \le \frac{M-2}{2n} \right\} . \tag{6}$$

Note that under $E_{\text{mod}}$, no modular wraparound occurs in the algorithm. Thus, for all valid levels $r$ and indices $j$, we have $\tilde{X}_i(r,j) = cX_i(r,j) + \xi_i(r,j)$ and

$$\hat{H}(r,j) = \sum_{i=1}^{n} \frac{\tilde{X}_i(r,j)}{c} = \sum_{i=1}^{n} \left( X_i(r,j) + \frac{\xi_i(r,j)}{c} \right) .$$

Next, we define the event

$$E_{\text{diff}} = \bigcap_{j=1}^{b} \left\{ \left| F_j - \hat{F}_j \right| \le \sqrt{2\sigma^2 n \log_2(b) \log(4b/\delta)} \right\} . \tag{7}$$

We will show later that $E_{\text{mod}}$ and $E_{\text{diff}}$ holds with high probability; for now, we assume that they hold.

**Privacy Analysis.** The privacy analysis follows similar to Theorem 1, with the difference that each hierarchical histogram has a total $2b$ entries (as opposed to $b$ for the flat histogram) and an $\ell_2$ sensitivity of $\log_2 b$ (as opposed to 1 for the flat histogram), since it contains one non-zero entry for each level of the tree.

**Utility Analysis.** We denote $F_j$ as $F(j)$ for ease of reading. Using the triangle inequality, we get,

$$\begin{aligned}
\Delta_p(\hat{F}, F) &= \left| F\big(j_p^*(\hat{F})\big) - p \right| \\
&\le \frac{1}{n} \left| F\big(j_p^*(\hat{F})\big) - \hat{F}\big(j_p^*(\hat{F})\big) \right| + \left| \frac{1}{n} \hat{H}\big(j_p^*(\hat{F})\big) - p \right| \\
&\le \max_{j \in [b]} \left\{ \frac{1}{n} \left| F(j) - \hat{F}(j) \right| \right\} + R_p^*(\hat{F}) .
\end{aligned}$$

The first term is bounded under $E_{\text{diff}}$ and this gives the utility bound.

**Bounding the Failure Probability.** Follows similar to Theorem 1 by using Claim 6 and Claim 7 instead of Claim 4 and Claim 5 respectively. □

We state and prove bounds on probabilities of the events $E_{\text{mod}}, E_{\text{diff}}$ defined above.

**Claim 6.** *If $M \ge 2 + 2cn + 2n\sqrt{2\sigma^2 \log(16nb/\delta)}$, then $\mathbb{P}(E_{\text{mod}}) \ge 1 - \delta/4$.*

*Proof.* Each discrete Gaussian random variable $\xi_i(r,j)$ is centered and sub-Gaussian with variance proxy $\sigma^2$ (cf. Property 8). A Cramér-Chernoff bound (cf. Lemma 10) gives us the exponential tail bound

$$\mathbb{P}\left( |\xi_i(r,j)| > \sqrt{2\sigma^2 \log(16nb/\delta)} \right) \le \frac{\delta}{8nb} .$$

Applying the union bound over $i = 1, \cdots, n$ and the $2b - 2$ nodes in each hierarchical histogram $x_i$ (each node corresponding to one $(r,j)$ pair) completes the proof. □

**Claim 7.** *We have $\mathbb{P}(E_{\text{diff}}|E_{\text{mod}}) \ge 1 - \delta/2$.*

*Proof.* Under $E_{\text{mod}}$, we have that $\hat{H}(j) = H(j) + \sum_{i=1}^{n} \sum_{(r,o) \in P_j} \xi_i(r,o)$, where $P_j$ is the maximal dyadic partition of $[1, j]$ with $|P_j| \leq \log_2 b$. Thus, $\zeta_j := \hat{H}(j) - H(j)$ is sub-Gaussian with variance proxy $n|P_j|\sigma^2 \leq n\sigma^2 \log_2 b$. A Cramér-Chernoff bound (cf. Lemma 10) gives us

$$\mathbb{P}\left(|\zeta_j| > \sqrt{2\sigma^2 n \log_2(b) \log(4b/\delta)}\right) \leq \frac{\delta}{2b} .$$

Applying a union bound over $j = 1, \cdots, b$ completes the proof. $\qquad\square$

## A.3 Useful Results

We state the sub-Gaussian property of the discrete Gaussians.

**Property 8.** *Let $\xi$ be distributed according to $\mathcal{N}_{\mathbb{Z}}(\mu, \sigma^2)$. Then, $\mathbb{E}[\xi] = \mu$. Furthermore, if $\mu = 0$, then $\xi$ is sub-Gaussian with variance proxy $\sigma^2$, i.e., $\mathbb{E}[\exp(\lambda\xi)] \leq \exp(\lambda^2\sigma^2/2)$ for all $\lambda > 0$.*

The distributed discrete Gaussian mechanism gets privacy guarantees by adding a sum of discrete Gaussian random variables. We give a bound on its privacy. The following lemma is due to [16].

**Lemma 9** (Privacy of Sum of Discrete Gaussians). *Fix $\sigma \geq 1/2$. Let $A : \mathcal{X} \to^d$ be a deterministic algorithm with $\ell_2$-sensitivity $S$ for some input space $\mathcal{X}$. Define a randomized algorithm $\mathcal{A}$, which when given an input $x \in \mathcal{X}$, samples $\xi_1, \cdots, \xi_n \sim \mathcal{N}_{\mathbb{Z}}(0, \sigma^2 I_d)$ and returns $A(x) + \sum_{i=1}^{n} \xi_i$. Then, $\mathcal{A}$ satisfies $\varepsilon^2/2$-concentrated DP with*

$$\varepsilon = \min\left\{ \sqrt{\frac{S^2}{n\sigma^2} + \frac{\psi d}{2}}, \frac{S}{\sqrt{n}\sigma} + \psi\sqrt{d} \right\} ,$$

*where $\psi = 10 \sum_{i=1}^{n-1} \exp\left(-2\pi^2\sigma^2 i/(k+1)\right) \leq 10(n-1)\exp(-2\pi^2\sigma^2)$.*

Next, we record standard concentration results (see e.g. [3]).

**Lemma 10** (Cramér-Chernoff). *Let $\xi$ be a real-valued and centered sub-Gaussian random variable with variance proxy $\sigma^2$, i.e., $\mathbb{E}[\xi] = 0$ and $\mathbb{E}[\exp(\lambda\xi)] \leq \exp(\lambda^2\sigma^2/2)$ for all $\lambda > 0$. Then, we have for any $t > 0$,*

$$\mathbb{P}(|\xi| > t) \leq 2\exp\left(-\frac{t^2}{2\sigma^2}\right) .$$

**Lemma 11** (Maximal Inequality). *Let $\xi_1, \xi_2, \cdots$ be i.i.d. centered sub-Gaussian random variables with variance proxy $\sigma^2$, i.e., $\mathbb{E}[\xi_j] = 0$ and $\mathbb{E}[\exp(\lambda\xi_j)] \leq \exp(\lambda^2\sigma^2/2)$ for all $\lambda \in \mathbb{R}$ and $j = 1, 2, \cdots$. Then, it holds for any $t > 0$ and integer $n \geq 1$ that*

$$\mathbb{P}\left(\max_{i \in [n]} \left|\sum_{j=1}^{k} \xi_j\right| > t\right) \leq 2\exp\left(-\frac{t^2}{2\sigma^2 n}\right) .$$

# B  DP-$\Delta$-FL: Background and Details

We describe the proposed DP-$\Delta$-FL approach, a variant of $\Delta$-FL [18] under end-to-end differential privacy.

## B.1  $\Delta$-FL Review

We review the $\Delta$-FL objective and the corresponding optimization algorithm [18].

Suppose we have $n$ clients, where the objective on client $i$, denoted $f_i : \mathbb{R}^d \to \mathbb{R}$, is the expected loss on client $i$ under its data distribution $q_i$ for $i = 1, \cdots, n$.

The $\Delta$-FL objective minimizes the tail mean of the per-client losses, which can be formalized using a notion called superquantile [22]. The $p$-superquantile of a continuous random variable $Z$ is defined as $\mathbb{S}_p(Z) = \mathbb{E}[Z \mid Z > Q_p(Z)]$, where $Q_p(Z)$ is the $p$-quantile of $Z$. More generally, the following definition is applicable to both discrete and continuous random variables:

$$\mathbb{S}_p(Z) = \min_{\eta \in \mathbb{R}}\left\{\eta + \frac{1}{1-p}\mathbb{E}\left[\max\{0, Z - \eta\}\right]\right\} .$$

---
**Algorithm 3** The $\Delta$-FL Algorithm [18]
---
**Input:** Initial iterate $w^{(0)}$, number of communication rounds $T$, number of clients per round $m$,
    number of local updates $\tau$, local step size $\gamma$
 1: **for** $t = 0, 1, \cdots, T - 1$ **do**
 2:      Sample $m$ clients from $[n]$ without replacement in $S$
 3:      Find the $p$-quantile of $\left(f_i(w^{(t)})\right)_{i \in S}$; call this $Q^{(t)}$
 4:      **for** each selected client $i \in S$ in parallel **do**
 5:         Set $\tilde{\pi}_i^{(t)} = \mathbb{I}\left(f_i(w^{(t)}) \geq Q^{(t)}\right)$
 6:         Initialize $w_{k,0}^{(t)} = w^{(t)}$
 7:         **for** $k = 0, \cdots, \tau - 1$ **do**
 8:            $w_{i,k+1}^{(t)} = (1 - \gamma\lambda)w_{i,k}^{(t)} - \gamma\nabla f_i(w_{i,k}^{(t)})$
 9:         **end for**
10:      **end for**
11:      $w^{(t+1)} = \sum_{i \in S} \tilde{\pi}_i^{(t)} w_{i,\tau}^{(t)} / \sum_{i \in S} \tilde{\pi}_i^{(t)}$
12: **end for**
13: **return** $w_T$
---

In this expression, the quantile $Q_p(Z)$ is obtained as the left end-point of the argmin set over $\eta$.

The $\Delta$-FL objective is to minimize the superquantile of the per-client losses:

$$g_p(w) := \mathbb{S}_p\left(f_1(w), \cdots, f_n(w)\right), \tag{8}$$

where $w \in \mathbb{R}^d$ denotes the model parameters.

**Algorithm.** The algorithm to optimize the $\Delta$-FL objective proposed by [18] can be interpreted more transparently as below. We take this alternate interpretation as opposed to the one used in [18] as it directly allows us to derive a version with end-to-end differential privacy.

The superquantile is a nonsmooth function, and so is the $\Delta$-FL objective $g_p$. However, leveraging duality, the following subdifferential expression can be derived.

**Proposition 12.** *Suppose the per-client objective $f_i$ is $L$-smooth for each $i$, and let the parameter $p$ be such that $np$ is an integer. For all $w$ such that $f_i(w) \neq f_j(w)$ for $i \neq j$, a subgradient of the objective* (8) *is given by*

$$\partial g_p(w) \ni \sum_{i=1}^{n} \pi_i^\star f_i(w), \quad \text{where} \quad \pi_i^* = \frac{\mathbb{I}(f_i(w) \geq Q_p)}{\sum_{j=1}^{n} \mathbb{I}(f_j(w) \geq Q_p)},$$

*and $Q_p = Q_p(f_1(w), \cdots, f_n(w))$ is the $p$-quantile of the losses and $\partial g_p$ refers to the regular subdifferential of $g_p$.*

*Proof.* Without loss of generality, we assume that $f_1(w) < \cdots < f_n(w)$. Due to the dual expression for the supqerquantile [11], we have the equivalent expression

$$g_p(w) = \max\left\{\sum_{i=1}^{n} \pi_i f_i(w) : 0 \leq \pi_i \leq \frac{1}{n(1-p)}, \sum_{i=1}^{n} \pi_i = 1\right\}. \tag{9}$$

By Danskin's theorem, we have that

$$\partial g_p(w) \ni \sum_{i=1}^{n} \pi_i^\star \nabla f_i(w), \tag{10}$$

where $\pi^\star$ attains the argmax in the expression above. Next, we invoke the a closed form expression of the superquantile for discrete random variables [23, Proposition 8] (this is analogous to the expression $\mathbb{S}_p(Z) = \mathbb{E}[Z \mid Z > Q_p(Z)]$ for continuous random variables $Z$) to get

$$g_p(w) = \frac{1}{(1-p)n} \sum_{i=i^\star+1}^{n} f_i(w) + \left(1 - \frac{\lfloor (1-p)n \rfloor}{(1-p)n}\right) f_{i^\star}(w),$$

where $i^\star = \lceil (1-p)n \rceil$. Comparing with (9), this gives a closed-form expression for $\pi^\star$, which is unique because $F_{i^\star-1}(w) < F_{i^\star}(w) < F_{i^\star+1}(w)$. By the definition of the quantile, also observe that $Q_p = f_{i^\star}(w)$. Plugging in this closed form expression of $\pi^\star$ into (10) completes the proof. $\qquad\square$

Using this expression, the algorithm of [18] interleaves federated averaging steps with quantile estimation. Specifically, the local updates $w_i^+$ from the subsample of $m$ selected clients $i \in S$ are aggregated to update the global model with the following two steps:

- compute $Q_p = Q_p(f_i(w) : i \in S)$,
- aggregate the updates from the tail clients where $f_i(w) \geq Q_p$ to find the new global model $w^+$ as

$$w^+ = \frac{1}{|S_p|} \sum_{i \in S_p} w_i^+ , \quad \text{where} \quad S_p = \{ i \,:\, f_i(w) \geq Q_p \} .$$

Similar to FedAvg, this aggregation rule enjoys a simplification in the case of a single local update per-client with a learning rate $\gamma$. Specifically, under the assumption of full client participation (i.e., $m = n$), if the local update $w - w_i^+ = \gamma \nabla f_i(w)$ is a single gradient step, the aggregated update is simply a subgradient step $w - w^+ = \gamma \nabla g_p(w)$ where we denote the subgradient as $\nabla g_p(w) \in \partial g_p(w)$.

The overall algorithm is summarized in Algorithm 3.

## B.2 End-to-End Differential Privacy with DP-$\Delta$-FL

To obtain an end-to-end differentially private version of $\Delta$-FL, we make two modifications to Algorithm 3. First, we estimate the quantile with distributed differential privacy, using either Algorithm 1 or Algorithm 2. Second, we modify the weight aggregation step (line 11) by clipping the weight updates and add Gaussian noise to obtain differential privacy via the Gaussian mechanism. The overall algorithm is given in Algorithm 4.

**Privacy Accounting.** We now discuss the privacy spent in each communication round. For simplicity, we assume the number $m^{(t)} = \sum_{i \in S} \mathbb{I}(f_i(w^{(t)}) \geq Q^{(t)})$ of selected clients is publicly known.

**Claim 13.** *Consider the setting of Algorithm 4 with noise scale $\sigma_w$, norm bound $C$ and Algorithm 2 with $b$ bins and noise scale $\sigma = \sigma_q$. Each round of Algorithm 4 satisfies $(1/2)\varepsilon^2$-concentrated DP where*

$$\frac{1}{2}\varepsilon^2 = \frac{1}{2}\varepsilon_q^2 + \frac{\sigma_w^2}{2C^2} ,$$

*where $\varepsilon_q$ is obtained from Theorem 1 or 2 for Algorithm 1 or 2 respectively.*

*Proof.* The $(1/2)\varepsilon_q^2$-concentrated DP of the quantile computation comes from Theorem 1 or 2. Since the contribution $\delta_i^{(t)}$ of each client has $\ell_2$ norm $\left\| \delta_i^{(t)} \right\| \leq C$ and we add Gaussian noise $\mathcal{N}(0, \sigma_w^2 I_d)$, the weight update step satisfies $\sigma_w^2/(2C^2)$-concentrated DP. The proof is completed by noting that concentrated differential privacy composes additively. $\qquad\square$

To obtain a bound on the concentrated DP of the entire algorithm, we rely on generic upper bounds of [27] for privacy amplification by subsampling.

## B.3 Experimental Setup

We consider a synthetic classification dataset and train a linear model on it.

**Dataset Description.** We create a 10-class classification dataset in $d = 20$ dimensions, inspired by [14]. The input $x$ for each class $k$ is drawn from a Gaussian of mean $\mu_i$ and identity covariance in $\mathbb{R}^{15}$. The means $\mu_i$'s are the corners of a random polytope in $\mathbb{R}^{15}$. We add 2 features that are linear combinations of the 15 informative ones and 3 features that are pure noise. Overall, the dataset can be generated using the `make_classification` function of scikit-learn [21] as

**Algorithm 4** DP-$\Delta$-FL: $\Delta$-FL with End-to-End Differential Privacy

---

**Input:** Initial iterate $w^{(0)}$, number of communication rounds $T$, number of clients per round $m$, number of local updates $\tau$, local step size $\gamma$, $\ell_2$ norm bound $C$ for weight updates, noise variance $\sigma_w^2$

1: **for** $t = 0, 1, \cdots, T-1$ **do**
2:      Sample $m$ clients from $[n]$ without replacement in $S$
3:      Estimate the $p$-quantile of $f_i(w^{(t)})$ for $i \in S$ with distributed differential privacy (Algorithm 1 or 2); call this $Q^{(t)}$
4:      Set $m^{(t)} = \sum_{i \in S} \mathbb{I}\left(f_i(w^{(t)}) \geq Q^{(t)}\right)$
5:      **for** each selected client $i \in S$ in parallel **do**
6:          Initialize $w_{k,0}^{(t)} = w^{(t)}$
7:          **for** $k = 0, \cdots, \tau - 1$ **do**
8:              $w_{i,k+1}^{(t)} = (1 - \gamma\lambda)w_{i,k}^{(t)} - \gamma\nabla f_i(w_{i,k}^{(t)})$
9:          **end for**
10:          Define the norm-clipped update contributed by the client

$$\delta_i^{(t)} = \begin{cases} \dfrac{C\left(w_{i,\tau}^{(t)} - w^{(t)}\right)}{\max\left\{C, \left\|w_{i,\tau}^{(t)} - w^{(t)}\right\|_2\right\}}, & \text{if } f_i(w^{(t)}) \geq Q^{(t)} \\ \mathbf{0}_d, & \text{else} \end{cases}$$

11:      **end for**
12:      Sample Gaussian noise $\xi^{(t)} \sim \mathcal{N}(0, \sigma_w^2 I_d)$ and update

$$w^{(t+1)} = w^{(t)} + \frac{1}{m^{(t)}}\sum_{i \in S}\delta_i^{(t)} + \xi^{(t)}$$

13: **end for**
14: **return** $w_T$

---

```
x, y = make_classification(
        n_samples=int(5e5), n_features=20,
        n_informative=15, n_redundant=2, n_repeated=0,
        n_classes=10, n_clusters_per_class=1,
        class_sep=5.0, hypercube=False, random_state=2345
)
```

We now split this dataset into a federated dataset with $n = 2500$ training clients and $n' = 500$ validation and $n'' = 500$ test clients. The data distribution $q_i(x, y) = q_i(y)q_i(x|y)$ across the clients is designed to exhibit a label shift, i.e., the distribution $q_i(y)$ over labels for each client is different while the class-conditional distribution $q_i(x|y = k) = \mathcal{N}(\mu_k, I_d)$ is the same across clients. The class distribution $q_i(y)$ on each training client $i$ is drawn from a Dirichlet distribution $\mathrm{Dir}(0.5)$, while that for a validation or test client is drawn from $\mathrm{Dir}(0.01)$. We sample 100 input-output pairs for each training, validation, and test client.

**Model and Per-Client Objective.** We use a linear model (with intercept) on each client and the multinomial logistic loss, also known as the cross entropy loss, to define the per-client objective.

**Algorithms and Hyperparameters.** We compare Algorithm 4 with DP-FedAvg [20], a version of FedAvg with differential privacy.

Both algorithms used a single full gradient step per client with a fixed learning rate of 0.1. For each algorithm, we sample 100 clients per round and run the training for a total of 1000 rounds. We vary the privacy budget $\varepsilon \in \{3, 5, 10, 15, 20\}$ and tune the following hyperparameters for each algorithm.

For DP-FedAvg, we tune the $\ell_2$ norm bound (analogous to $C$ in Algorithm 4) and set the noise scale $\sigma_w$ depending on the target privacy budget $\varepsilon$ and the norm bound $C$. For Algorithm 4, we allocate $r$-times the privacy budget of the weight updates to the quantile updates. In addition, we also tune:

- the loss upper bound $B$, so that all losses are truncated to $[0, B]$,
- the number of bins $b$ in the hierarchical histogram,
- the $\ell_2$ norm bound $C$ for the weight update.

We tune all $4$ hyperparameters with a grid search and set the noise scale $\sigma_w$ for the weight update, and $\sigma_q/c$ for the quantile update depending on the selected hyperparameters and the privacy budget $\varepsilon$. The objective of the grid search was to minimize the $90^{\text{th}}$ percentile of the misclassification errors across all validation clients.

The ranges of the hyperparameters considered are quantile privacy ratio $r \in \{0.1, 0.25, 0.5, 0.75\}$, loss upper bound $B \in \{0.7, 0.9, 1.1, 1.3, 1.5\}$,[2] number of bins $b \in \{16, 32, 64\}$, and update norm $C \in \{0.9, 1.1, 1.3, 1.5\}$.[3]

---

[2]The loss at convergence was around $0.7$, while that at random guessing is $\log 10 \approx 2.3$.

[3]These correspond approximately to the $0.3, 0.5, 0.7, 0.9$ quantiles of the update norms of FedAvg without differential privacy, during the latter half of training.

