# OpenReview forum: "Differentially Private Federated Quantiles with the Distributed Discrete Gaussian Mechanism"
_NeurIPS.cc/2022/Workshop/Federated_Learning — FL-NeurIPS 2022 Poster_

### Official Review · Reviewer_Kdo1 · 2022-10-09

Summary:
This paper proposes two quantile estimation algorithms based on distributed discrete Gaussian mechanism with the consideration of secure aggregation, and present utility and privacy analysis and comparisons of the two approaches with numerical experiment results. The algorithms are also applied to distributionally robust federated learning.

Strengths:
First, this paper focuses on the federated quantile computation of per client, which is essential for robust statistics and adaptive clipping in federated averaging, also for distributionally robust versions. The previous approaches are still not clearly to be extended to satisfy differential privacy with limited communication resource, and this paper proposes an simple alternative method to be compatible with distributed differential privacy.
The writing is mostly clear, and their theory and experiments support their arguments well with mathematical conductions, and they also properly cite the other papers.

Weaknesses:
The potential concern of computational and communication cost in federated learning setting has not been considered. Each client needs to compute the hierarchical histograms, which may not be fulfilled with limited computational resources and batteries. Uploading the local histogram may also introduce communication overhead. They may also add more illustration on the datasets and network models they use for the client vectors.

---

### Official Review · Reviewer_2oD3 · 2022-10-18

Overall, this paper has high quality.
**Clarity**
The overall clarity is great. One minor issue: $\mu$ is not defined in the paragraph "Quantiles from Flat Histograms".
**Originality**
This paper provides two novel quantile computation algorithms in the setting of federated learning. The theoretical analysis for privacy and utility looks sound.
**Significance**
The application of quantile estimation to distributionally robust federated learning looks very promising. However, I am not very familiar with the literature.

---

### Decision · Program_Chairs · 2022-10-20

Accept (Poster)